# The impact of COVID-19 on Physical Activity of Czech children

**Tereza Štveráková** [1]*, **Jakub Jačisko**[1], **Andrew Busch** [2], **Marcela Šafářová**[1], **Pavel Kolář**[1], **Alena Kobesová**[1]

**1** Department of Rehabilitation and Sports Medicine, Postgraduate Medical School, Second Faculty of Medicine Charles University and Motol University Hospital, Prague, Czech Republic, **2** Health and Human Kinetics, Ohio Wesleyan University, Delaware, OH, United States of America

* stverkaterka@gmail.com

## Abstract

### Introduction

The pandemic of coronavirus disease (COVID-19) and related restrictions (closed schools and sports centers, social isolation, masks) may have a negative impact on children's health. The purpose of this study was to evaluate the level of physical activity (PA) of Czech children during COVID-19 in autumn 2020.

### Methods

Ninety-eight Czech children (mean age = 10.1 ± 1.47 years) completed the standardized Physical Activity Questionnaire for Older Czech Children (PAQ-C/cz) during COVID lockdown. Data were compared with previously published norms. Thirty-five children also reported daily number of steps measured by accelerometers.

### Results

Total PAQ-C score was 0.38 lower during COVID compared to Pre-COVID [$t$(302) = 5.118., $p < .001$]. The male PAQ-C total score was 0.37 lower [t(146) = 3.21., p = .002)] and the female total score was 0.39 lower [t(154) = 3.97., p < .001] during COVID compared to Pre-COVID. Specifically, responses of PA during spare time, before-school, physical education (PE), and recess were significantly lower during COVID. The average number of steps was 7.767 steps/day (boys = 9.255; girls = 6.982).

### Conclusion

COVID lockdown resulted in significant reduction of PA in Czech children. Strategies to promote adequate PA of children during the pandemic need to be determined.

**Citation:** Štveráková T, Jačisko J, Busch A, Šafářová M, Kolář P, Kobesová A (2021) The impact of COVID-19 on Physical Activity of Czech children. PLoS ONE 16(7): e0254244. https://doi.org/10.1371/journal.pone.0254244

**Data Availability Statement:** All relevant data are within the paper and its Supporting Information files. The supporting information is included at the end of the manuscript on pages 19-20, lines 397-401. Supplementary Materials: available online at:

https://journals.plos.org/plosone/article?id=10.
1371/journal.pone.0245256.

**Funding:** This study was supported by
Rehabilitation Prague School (www.rehabps.com),
by Institutional research program Progress Q41
and by the foundation Movement without Help.

**Competing interests:** The authors have declared
that no competing interests exist.

## Introduction

Healthy physical development in children is largely dependent on sufficient physical activity
(PA), reduced sedentary behavior (SB) and adequate sleep. These three factors are referred to
as a movement behavior [1]. According to the World Health Organization (WHO), lack of
regular PA and increased time spent in sedentary activity are globally the fourth highest risk
factor attributed to mortality, with overweight and obesity being the third leading risk factor of
mortality in middle and high-income countries, behind only high blood pressure and tobacco
use [2, 3].

Regular PA promotes general health, prevents obesity and other civilization diseases [3]. To
meet the criteria of the optimal movement behavior, it is recommended that children and ado-
lescents aged 5-13 years strive to achieve a daily minimum of 60 minutes of moderate to vigor-
ous PA, limit sedentary recreational screen time to 2 hours maximum, and acquire 9 to 11
hours of uninterrupted sleep per night [1, 4].

Beginning around six years of age, children undergo significant psychosocial developmental
changes reflecting their self-concept and relationship with the environment [5]. An important
part of their daily routine and healthy lifestyle is their regular school attendance and organized
PA [5–8]. It contributes to the improvement of social contacts and can influence quality of life.
There is also strong evidence promoting the effectiveness of regular PA and exercise in the
treatment of depression, anxiety and improving mental well-being [9].

Many countries have successfully worked towards these goals of optimal children's move-
ment behavior by implementing different organized sport activities and integrating regular
physical education (PE) lessons in schools. However, in December 2019 the first cases of the
coronavirus disease 2019 (COVID-19) were reported and on January 30 COVID-19 became
officially declared international public health emergency [10]. Governments implemented
restrictions involving school and sport grounds closures resulted in health risks behaviors espe-
cially reduced PA and increase SB [11]. The regulations had negative effect on various mental
and health aspects in children and youth such as increasing obesity [12, 13], pain [14], depres-
sion, anxiety, loneliness feelings [15–17], sleep disturbances [18, 19], decreased cardiorespira-
tory fitness [20] and many others, affecting especially socio-economic deprived children [11].

PA in Czech children and youth was reported to be insufficient already before the COVID.
Only 35% of children population performed the recommended amount of PA, i.e. 60 minutes
of moderate to vigorous PA per day. The high rates of excessive screen-time were reported by
Gaba et al. in study collecting data during the pre-pandemic 2018 year. Organized PA and
sport have been performed by 55% of girls and 70% of boys, joint sports activities with the fam-
ily at least once a week were reported only by 34% of girls and 37% of boys. Under normal con-
ditions all schools in the Czech Republic must guarantee at least 90 min per week of PE [21].
However, this amount of time is considered insufficient and there has been a long-standing
political and professional discussion on increasing PE classes at primary schools. Most schools
offer more PE classes than the mandatory 90 minutes per week and provide favorable environ-
ment to promote PA outside PE classes. Still, there has been a call for increase of PE hours in
elementary schools and after-school health-enhancing PA promotion [21, 22].

The first three cases of COVID-19 were reported in the Czech Republic on March 1st, 2020
[23]. Outbreak of COVID-19 resulted in full shutdown of organized sports and public sports
facilities in the Czech Republic. From March 2020 till May 2021 school attendance and orga-
nized sport activities have been largely unavailable in the Czech Republic. By April 2021, the
Czech Republic has recorded the second highest confirmed death rate in the world with 1.8%
case fatality and 282.14 deaths per 100 000 people [24]. With just a shortbreaks, children were
ordered to stay at home, online education was established, and organized sport activities were

prohibited both indoor and outdoor. Such long lasting restrictions may create unintended poor habits of decreased PA and increased SB in child populations [25], having negative effects on daily routines and opportunities for being active [26].

Movement activities of children and adolescents during COVID differ among countries due to different policy restrictions and the number of COVID-19 infections [26]. According to UNESCO, from all European Union (EU) countries, the Czech Republic had the longest school closure during the pandemic, that is 42 weeks [27]. Even countries with much shorter period of school closure, such as France (11 weks school closure) [28] or Portugal (24 weeks closure) [29] or Spain (15 weeks closure) [30] confirmed decreased levels of PA in children during COVID calling for the development of effective national action. German study where the schools were closed for 30 weeks reports decreased sports activity but increase in habitual physical activities such as gardening, housework, cycling or walking [26]. Other studies also report the shift of PA towards nonorganized outdoors activites such as walking, running, bycycling and alike [19, 31]. This study attempts to analzye the impact COVID-19 has on PA on children in the Czech Republic, where COVID death rates were very high and the schools and sport faciles closed for the longest period of time in the whole EU.

It is difficult to determine the most appropriate tools to evaluate physical behavior aspects in children and the "gold standard" is still not available [4, 32–34]. Despite a certain degree of bias, questionnaires and accelerometers are currently the most widely used methods to collect such data. The principle of self-assessment questionnaires is based on the respondent's ability to recall his or her activities during the observed period of time, usually one week or one month back. Although only some questionnaires demonstrate adequate validity and reliability, they represent a cheap and easy way to assess the amount of PAs [1, 35, 36].

The Physical Activity Questionnaire (PAQ) is one of the most frequently used questionnaires worldwide [37]. The Physical Activity Questionnaire for Older Children (PAQ-C) variant was developed for children aged 8–14 years old [32] and was recently standardized for Czech children by Cuberek et al. [38]. PAQ-C is considered a reliable tool to evaluate children's PA [5, 38–40]. Questionnaires can be combined with data obtained from accelerometers that monitor the amount of PA. Although accelerometers do not capture certain types of PA accurately (such as cycling), they can provide a rough estimate of the level of PA achieved by the subject throughout the day [35].

This is the first study comparing PA of school children during the COVID lockdown time with pre-COVID norms defined by the Czech version of PAQ-C (PAQ-C/cz) and by evaluating their number of daily steps.

## Methods

### Study design

An anonymous survey to evaluate PA of children aged 8–12 years was conducted during COVID Lockdown in the Czech Republic in November and December 2020. The data were compared with the Pre-COVID norms defined by the same questionnaire (PAQ-C/cz) (see the S1 and S2 Files). The subjects were also asked to monitor and report the number of daily steps if they have appropriate measurement device (smart watch or smart phones) available. The study was approved by the Ethics Committee of the University Hospital Motol and 2nd Faculty of Medicine, Charles University in Prague (EK-1730/20).

### Participants

Our data were collected during COVID lock down from November 2020 to January 2021. It was compared with the norms collected by Cuberek et al. [38] one year earlier, during the

same time of the year before the COVID-19 occurred in the Czech Republic. Participants were recruited using an information leaflet created for the purpose of this study. It was published either electronically on an official Dynamic Neuromuscular Stabilization (DNS) website www.rehabps.com or in paper form at physiotherapy centers and at the Motol University Hospital. It contained information on the purpose of the study, a standardized questionnaire and the informed consent. Participation in the study was voluntary and the informed consent was signed by the participant's parent or legal representative. In total, ninety-eight children participants (56 girls and 42 boys) completed the questionnaire either electronically or as a hard copy. Inclusion criteria for participation comprised age (8–12 years) and participation in distance learning education. Participants were excluded if they had any serious health condition. The questionnaire data revealed participants in this cohort (98), were living in cities and villages of different sizes, came from different elementary schools, and came from various family and social backgrounds. Table 1 shows anthropometric characteristics of the study participants.

## Physical Activity Questionnaire for Older Children (PAQ-C/cz)

A validated Czech version of the standardized PAQ-C/cz was used to evaluate the level of PA in the observed cohort. The standardized PAQ-C was recently adapted into the Czech version by Cuberek et al., which assessed its psychometric properties and recommended it as a tool for physical activity assessment in large-sample research studies [38].

PAQ-C/cz is a ten-item, self-administered, seven-day recall questionnaire for children 8–14 years old. The questionnaire provides a summary of PA calculated from nine items, each scored on a five-point scale (with 1 representing the lowest level and 5 representing the highest level of PA). The total PAQ-C score is calculated as a mean value from the nine different item scores including: children´s spare time activity (question 1), activity before school (question 2), activity during physical education (PE) lessons (question 3), activity during recesses (question 4), activity after school (question 5), activity in the evening (question 6), activity during weekend (question 7), statement of free time activity during last week (question 8), and activity level and frequency performed each day during last week (question 9). Question number 10 is of a qualitative character inquiring about any disease or other obstacles to perform PA during the observed period of time and therefore cannot be included in the final score calculation [38].

Because the survey was done at a time of distance online school education, there was a note in the questionnaire form to evaluate PA during recesses as the time between online lessons. Collected COVID data were compared with the Pre-COVID norm data recently published by Cuberek et al. [38]. See Table 1 to compare demographic characteristics of COVID and Pre-

**Table 1. Study characteristics comparing Cuberek et al. [38] Pre-COVID data with during COVID data of Czech children.**

|  | Sex | Sample Size | Age* | BMI* |
|---|---|---|---|---|
| Cuberek et al Pre-COVID cohort | Male | 106 | 11.08 (0.84) | 18.46 (3.13) |
|  | Female | 100 | 11.17 (0.82) | 17.36 (2.68) |
|  | Total | 206 | 11.13 (0.83) | 17.92 (2.97) |
| During COVID cohort | Male | 42 | 10.21 (1.49) | 17.56 (3.06) |
|  | Female | 56 | 10.02 (1.46) | 17.20 (2.70) |
|  | Total | 98 | 10.10 (1.47) | 17.35 (2.85) |

*Reported as mean (standard deviation), note: BMI: Body mass index.

COVID cohorts. Additionally, thirty-five children from our COVID cohort reported daily number of steps using smart watch or smart phones to count the steps. Children provided a print screen from the device to prove the number of steps for each day.

## Statistical analysis

Descriptive statistics were calculated for all variables. Data are mean ± standard deviation, unless otherwise stated. Independent-samples t-tests (2-tailed) were performed to assess differences in PA from the PAQ-C scores among Czech children between the period of COVID lockdown with published PA norms prior to COVID restrictions. Statistical significance was determined a priori at $P < 0.05$ for the PAQ-C total score. When comparing responses to individual questions within the PAQ-C, Bonferroni corrections were utilized to reduce chances of Type 1 error, and was set at $P < 0.005$. Power analysis, using G*Power 3.1, indicated 128 subjects were needed (64 per group) to detect a medium effect size of 0.5 and an achieved power of 0.80. Effect sizes were interpreted as very small ($< 0.2$), small ($0.2–0.5$), medium ($0.5–0.8$), or large ($> 0.8$) [41]. Data analyses were conducted using the Statistical Package for the Social Sciences v27 (SPSS Inc, Chicago, IL).

## Results

Distribution of the 98 participants during COVID lockdown were: males (n = 42, 42.9%), females (n = 56, 57.1%) and the 206 participants included from pre-COVID data were: males (n = 106, 51.5%), females (n = 100, 48.5%). Participant characteristics for both COVID lockdown and pre-COVID data are outlined in Table 1. Not all data was normally distributed, as assessed by Shapiro-Wilk's test. Due to the robustness of the independent samples t-test, data was not altered. Cronbach's alpha scores were calculated to score internal consistency for both sets of PAQ-C/cz questionnaire data (pre-COVID and COVID lockdown) using all nine questions. Cronbach's alpha for pre-COVID questionnaire data (Cuberek et al. [38]) was acceptable at 0.758, and COVID lockdown Cronbach's alpha was interpreted as good at 0.806 [42]. Results of all independent samples t-tests with 95% confidence intervals are presented in Table 2, with gender-specific data presented in Table 3. Significant differences were found in

**Table 2. Comparison of Czech children scores on the PAQ-C regarding PA before and during COVID pandemic (mean [standard deviation]).**

| Measure | Cuberek et al. Pre-COVID (n = 206) | COVID Lockdown (n = 98) | Mean Difference (95% CI) | Effect Size | P Value |
|---|---|---|---|---|---|
| Total PAQ-C Score | 2.69 (0.59) | 2.30 (0.66) | 0.38 (0.24, 0.53) | 0.63 | < .001* |
| Q1 Spare time activity | 1.34 (0.22) | 1.26 (0.17) | 0.07 (0.03, 0.12) | 0.38 | .001** |
| Q2 Before-school activity | 2.06 (1.37) | 1.63 (1.08) | 0.43 (0.15, 0.72) | 0.34 | .003** |
| Q3 Physical education | 3.83 (1.15) | 2.26 (1.37) | 1.57 (1.26, 1.89) | 1.28 | < .001** |
| Q4 Recesses | 2.82 (0.95) | 1.87 (1.03) | 0.95 (0.71, 1.18) | 0.97 | < .001** |
| Q5 After-school activity | 3.00 (1.11) | 3.14 (1.19) | -0.14 (-0.42, 0.14) | -0.12 | 0.32 |
| Q6 Evenings | 2.59 (1.07) | 2.43 (1.30) | 0.16 (-0.11, 0.44) | 0.14 | 0.25 |
| Q7 Weekend | 2.90 (0.98) | 2.82 (0.92) | 0.08 (-0.15, 0.31) | 0.09 | 0.49 |
| Q8 Statement | 2.71 (1.04) | 2.65 (1.10) | 0.06 (-0.19, 0.32) | 0.06 | 0.64 |
| Q9 Weekly activity | 2.93 (0.76) | 2.70 (0.85) | 0.23 (0.04, 0.43) | 0.30 | 0.016 |

*Statistically significant difference observed ($P < 0.05$).

**Statistically significant difference observed (Bonferroni correction $P < 0.005$).

Note: PAQ-C: Physical Activity Questionnaire for Older Children.

Values are tabulated scores from PAQ-C.

Effect size = calculated Cohen's d.

**Table 3. Gender specific scores on the PAQ-C before and during COVID pandemic mean [standard deviation]).**

| Measure | | Cuberek et al. Pre-COVID | COVID Lockdown | Mean Difference (95% CI) | Effect Size | *P Value* |
|---|---|---|---|---|---|---|
| Total PAQ Score | Male | 2.69 (0.62) | 2.32 (0.69) | 0.37 (0.14, 0.61) | 0.59 | .002* |
| | Female | 2.68 (0.56) | 2.29 (0.64) | 0.39 (0.20, 0.58) | 0.66 | < .001* |

*Statistically significant difference observed (Bonferroni correction $P < 0.025$).

Note: PAQ-C: Physical Activity Questionnaire for Older Children.

Values are tabulated scores from PAQ-C.

Effect size = calculated Cohen's d.

the mean PAQ-C total scores between pre-COVID and COVID lockdown, $t(302) = 5.118$., $p < .001$, $d = .63$, with a mean difference of .385 (95% CI: .237, .532).

After a Bonferroni correction, independent samples *t*-tests compared answers on nine individual questions of the PAQ-C. Significant differences between pre-COVID and COVID lockdown mean scores were noted for: Spare time (Q1) $t(239.2) = 3.39$., $p = .001$, $d = .38$, before school (Q2) $t(236.9) = 2.97$., $p = .003$, $d = .34$, PE (Q3) $t(164.87) = 9.85$., $p < .001$, $d$- = 1.28, and recesses (Q4) $t(302) = 7.91$., $p < .001$, $d = .97$. No significant differences were noted for: After school (Q5) $p = 0.32$, evenings (Q6) $p = 0.25$, weekend (Q7) $p = 0.49$, statement (Q8) $p = 0.64$, or weekly activity (Q9) $p = 0.16$. See S1 Graph. There were no differences noted between genders when comparing PAQ-C total scores pre-COVID or during COVID lockdown. After dichotomizing COVID lockdown data into younger (8–9 yr., n = 43) and older (10–12 yr., n = 55) groups, no differences were noted between PAQ-C total scores ($p = .217$).

## Discussion

The COVID-19 pandemic has created unprecedented situations by which strict community lockdowns have negatively affected the movement and health behavior in children [11]. Negative consequences like increasing obesity [12, 13], pain [14], depression, anxiety, feelings of loneliness [15–17], and sleep disturbances [18, 19] are closely related to PA levels [31, 43, 44]. Our findings are similar to others [28, 30, 45] demonstrating a significant decline in PA in children during COVID compared to pre-COVID. This problem is quite complex however, due to the multifactorial nature of various changes in PA during COVID such as: social status and family income [19], city dwelling versus villages or suburbs [44], parental education levels [18], pre-pandemic sports habits [46], the level of national restrictions, and age [26] among others.

Studies suggest of a relationship between age and the amount of PA during lock down [26, 30]. Speculating that older children could be more prone to online games, watching movies and following the social networks than younger children, we compared the COVID data for younger (8–9 yrs, n = 43) and older (10–12 yrs, n = 55) groups, but no differences were noted between PAQ-C total scores. Today, unorganized free-play activities have become less common and children's time has been increasingly devoted to organized PA [47]. Due to COVID-19 regulations, even the youngest school children significantly reduced their PA and did not replace the regular amount of movement with any alternative PA such as playing in the garden, in the park, or running around the house. This was true both for the boys and the girls. We have not confirmed the increase in habitual outdoor PAs like other studies [26, 44]. A recently published study by Ng et al. on Czech adolescents' remote school and health experiences during spring lockdown reports more PA. This discrepancy is perhaps due to age differences since our study evaluates PA in children 8–12 years old, while the study by Ng was done on older children aged 11–15 years at a different time of year. We collected data in November-January,

while Ng collected their data from May-June, when more individual outdoor PA could be expected [48].

Organized sports of all kinds, both indoor and outdoor were significantly reduced in our respondents (Q1-spare time activity) as well as morning, i.e. before school activities (Q2). This is not surprising, since children did not walk to school and could not participate in regular sports because playgrounds and sports clubs were closed. The results of our survey also show that PE activities (Q3) were significantly limited as well. This should be considered and possibly changed by the PE teachers. If other school subjects can be taught online, there is no reason why PE could not. We can assume that almost every child can do simple exercises such as jumps, push-ups, sit-ups, plyometric exercises at home under the online guidance of the teacher. At the same time the PE teacher can motivate children and request to do individual activities such as running, walking, nordic-walking, scootering, bicycling and alike recording the frequency and intensity in a PA diary that should be signed by the parents and regularly presented to PE teacher. The same is true for the PA during recesses which were also reported significantly low. PE teachers could possibly take over during recesses to guide children through simple stretching exercises and repetitive aerobic movements under their online guidance to compensate for the SB. Strategies and recommendations for PE via distance learning have already been discussed in the literature confirming PE teachers' critical role in supporting student health during the COVID-19 pandemic [49] but the conditions differ significantly by country and local policies [50]. Unfortunately, this type of regime was not established in many Czech schools. Children in the questionnaire mostly responded "I did not have a PE lesson/I did not do PE". Also, the Czech pre-COVID PE score was rather low compared to other countries including data from Turkey, Great Britain, and China. In Turkey it was $4.52 \pm 0.99$ [51], two surveys in Great Britain reported a PE score of $4.14 \pm 0.80$ and $4.18 \pm 0.74$ [52] and in China $4.04 \pm 0.98$ [53]. The Czech pre-COVID PE score was $3.83 \pm 1.15$ and during COVID it was only $2.26 \pm 1.37$ [38].

There were no significant differences in after-school activities, evening, or weekend activities. We speculate this results from parental care motivating children and establishing routines for movement at a time when the family is together [30]. The most frequently reported PA during the period of quarantine restrictions was tourism and walking, followed by bicycling and athletics, specifically running (see S2 Graph). The weekly activities mean score was lower during-COVID compared with pre-COVID, but not enough to be statistically significant. The most active weekday was Saturday for both males and females, which is in line with other studies underlying the importance of accessibility to outdoor spaces for sufficient PA during the pandemic [19, 26, 44].

Tourism has a strong tradition in the Czech Republic, and likely represents the main type of regular PA during pandemic both for children and adults. COVID-19 associated regulations may change the structure of general population PA preferring the outdoor PA. Tourism is an optimal form of PA for the whole family and parental support is an important correlate of children's PA [19, 54]. The data from PAQ-C are in line with data obtained from the thirty-five subjects from our cohort who also reported the number of daily steps measured by the pedometers (smartphones, watch). The average number of daily steps was 7.767 steps with boys reporting 9.255 steps per day and girls 6.982 per day on average. The highest number of steps (10.244 steps on average) was measured on Saturday. Still, Czech children during COVID-19 do not meet the recommendations for the number of daily steps. Suggested number of steps for normal 6–12 years children population to maintain good health ranges from 12.000 to 16.000 steps/day [55–57]. For effective reduction of childhood obesity, the girls are recommended to take at least 11.000 and the boys at least 13.000 steps/day five days per week at minimum [58]. According to Vuković et al., children who were physically active before the

pandemic tend to continue their activities during the emergency state [46]. The insufficient amount of PA of Czech children before COVID became even more pronounced during COVID [21].

When comparing pre-COVID PAQ-C scores of Czech children with children of different countries, several differences exist. A Turkish study applied the PAQ-C survey to 784 primary school students (ages 9-14 years) and reported total PAQ-C scores to be 3.16 ± 0.73 [51]. A study in the United States performed the PAQ-C survey in a group of 1,172 children and noted differences when separated by race: European-American (3.36 ± 0.80), African-American (3.37 ± 0.69), and Hispanic (3.19 ± 0.64) [59]. Two British studies reported mean PAQ-C scores of 3.49 ± 0.68 (n = 336) and 3.36 ± 0.67 (n = 131) [52]. For Chinese children (n = 742), the total PAQ-C score was lower, 2.62 ± 0.68 [53]. The reported total PAQ-C score for the pre-COVID Czech population was only 2.69 ± 0.59, which means Czech children move less than Turkish, US and British children. Only Chinese children move slightly less then Czech. This is an alarming finding that even under normal conditions Czech children do not move sufficiently. The current findings of this study demonstrate a decrease in an already rather sedentary population of Czech children, which can only worsen as COVID lockdowns prolong.

Sufficient PA is critical in civilization disease prevention [60–62]. Children should be physically active daily as part of play, games, sports, transportation, recreation, PE, or planned exercise in the context of family and if possible in the context of school and community [62]. It seems that most families tried to compensate the lack of PA during COVID lockdown by tourism, especially on weekends. However, walking can be effective compensation only if optimal duration, speed, frequency, cardiorespiratory level, postural-stabilization and other parameters are respected [61, 62]. Especially the gait duration, speed and country terrain (hilly versus flat) is critical for sufficient oxygen uptake and aerobic fitness. For health benefits school-aged children and youth should accumulate at least 60 min of moderate to vigorous PA on a daily basis [61–63]. More daily PA provides greater health benefits [62]. To meet such criteria, brisk walking, jogging or hiking in nature is a good variant [61, 64]. We do not know the parameters of the reported tourism, and therefore cannot tell if it was an effective compensation for PA.

When analyzing both during-COVID and pre-COVID PA, the aspect of weather should be taken into account. The comparison of our data collected during COVID lockdown was coincidentally collected during the same months (November—January) as the pre-COVID data reported by Cuberek et al. just one year prior [38]. Rain, temperature, and earlier times of dusk may discourage children from doing outdoor activities. The autumn season is characterized by a decrease in energy expenditure in children attaining lower numbers of steps per day [65]. Perhaps spring and summer lockdowns would have less significant effects on children's PA.

Studies mapping the level of PA during COVID time in other countries exist, but other methods than PAQ-C were applied. An American study monitored the time spent by eleven common types of PA (walking, running, swimming, etc.) and twelve common types of SB (watching television, playing computer games, reading, etc.). The most common types of PA during the early-COVID-19 period was unorganized play and unstructured activities such as running around, hide and seek and similar games (90% of children) or going for a walk (55% of children). Parents of older children (9-13 years) admitted greater decreases in PA and greater increases in SB than parents of younger children (5–8 years) [66]. This was not confirmed by our study, because no differences were noted between younger and older children PAQ-C total scores (*p* = .217). The Canadian online study with children aged 12–17 evaluated PA, SB and sleep time during the March 2020 COVID-19 pandemic. Canadian children and youth had lower PA levels, less outdoor time, higher SB (including leisure screen time), and more sleep during the outbreak [67]. A Portuguese anonymous online survey examined

children aged up to 12 years at the end of March 2020. During COVID boys and girls performed PA equally but children with a previous routine of outdoor activities and children with siblings were more active. However, the total time spent being PA during COVID-19 was lower compared to normal days [45]. A significant reduction of PA during COVID is also reported in a Brazilian study [5], Spanish online survey [68] and Chinese study [69] using the International Physical Activity Questionnaire Short Form (IPAQ-SF) and the Profile of Mood States (POMS).

To our knowledge, this is the first study using the standardized PAQ-C to compare PA pre-COVID and during-COVID lockdown. The PAQ-C/cz questionnaire was recently validated [38] and the pre-COVID raw data were compared with during-COVID raw data collected during the same time of year (November-January). However, a limitation of PAQ-C is that the questionnaire does not offer detailed information about the intensity and time engaged in PA. Therefore, we combined the PAQ-C data with the number of steps reported by 35 subjects who had pedometers available. The use of pedometers is historically the oldest but still currently the most widespread way of instrumental PA monitoring [6, 70]. It is the suggested method to monitor PA to follow prescribed public health guidelines [71]. Although boys reported higher numbers of daily steps, due to the small sample size comparing only 12 boys with 23 girls we have identified no statistical difference between boys and girls. However, this trend is similar to normative data. Simply comparing the current dataset of Czech children's steps/day with previously published normative data, large differences are noted, which are concerning. It is typically noted that the number of steps/day peak before the age of 12 and slowly decrease throughout adolescence to approximately 8.000–9.000 steps/day by 18 years old. In children, boys typically average 12.000–16.000 steps/day, whereas girls average 10.000-13.000 steps/day [56]. The limited cohort in this study reported boys averaged 7.768 steps/day, and girls averaged only 6.982 steps/day. This concerning trend requires further investigation.

There are some limitations to this study. Only 98 children completed the questionnaire with only 35 also reporting the number of daily steps using smart watch/phones. Employment of such devices may have represented a certain motivation for children to take a larger number of steps. We expect that these children walked more than the rest of the cohort. Therefore, the average number of the steps in the whole cohort was most likely smaller then reported above. The data collection started at the time when some outdoor organized sport activities were still allowed (early November) while children who completed the survey later in December were protected from all organized sport activities. So, during the time of data collection the restriction orders kept slightly changing. This could possibly affect results. Another limitation could be the fact that the study may not fully represent the population as a whole. Parents who are not upset by the lockdown are perhaps not as motivated to complete a questionnaire regarding their child's lack of PA.

The authors of the study encourage researchers from other countries to use the internationally standardized PAQ-C to conduct surveys in their countries and compare the results internationally, to help establish optimal strategies for preventing detrimental effects of long lasting hypomobility in school-aged children.

## Conclusions

The "second wave" of the COVID-19 pandemic restrictions had a negative impact on PA of Czech boys and girls 8–12 years old. Based on comparison of Czech and international PAQ-C data, it seems that even under normal conditions Czech children are less physically active than their peers abroad. Further significant reduction of children's PA due to epidemic restriction is alarming. This topic should be considered a public health concern. School, sport and

government authorities need to set up effective strategies promoting school children's PA both during and after COVID.

## Supporting information

**S1 Table. Study characteristics comparing Cuberek et al. [38] Pre-COVID data with during COVID data of Czech children.**
(PDF)

**S2 Table. Comparison of Czech children scores on the PAQ-C regarding PA before and during COVID pandemic (mean [standard deviation]).**
(PDF)

**S3 Table. Gender specific scores on the PAQ-C before and during COVID pandemic (mean [standard deviation]).**
(PDF)

**S1 File. Master data: Pre-COVID and COVID.**
(XLSX)

**S2 File. The Physical Activity Questionnaire for Older Children—the Czech version.**
(PDF)

**S3 File.**
(DOCX)

**S1 Graph. Comparation of PAQ-C/CZ questionnaire results between Pre-COVID (n = 206) and COVID data (n = 98).** PAQ-C–total PAQ-C score; Q1—Spare time activity; Q2—Before school activity, Q3—Physical education; Q4—Recesses; Q5—After school activity; Q6 –Evening activity; Q7 –Weekend activity; Q8 –Statement; Q9 –Weekly activity.
(TIF)

**S2 Graph. Spare time activity during COVID lockdown–reported mean values for each type of PA (n = 98), frequency of individual sort of PA from question 1 (evaluation of these leisure activities is a standard part of PAQ-C).**
(TIF)

## Acknowledgments

The authors would like to thank all the individuals who volunteered their time to participate in our study.

## Author Contributions

**Conceptualization:** Tereza Štveráková, Jakub Jačisko, Andrew Busch, Marcela Šafářová, Pavel Kolář, Alena Kobesová.

**Data curation:** Tereza Štveráková, Jakub Jačisko, Andrew Busch.

**Formal analysis:** Tereza Štveráková, Jakub Jačisko, Andrew Busch, Alena Kobesová.

**Funding acquisition:** Alena Kobesová.

**Methodology:** Tereza Štveráková, Jakub Jačisko, Andrew Busch, Alena Kobesová.

**Resources:** Tereza Štveráková, Andrew Busch.

**Software:** Andrew Busch.

**Supervision:** Alena Kobesová.

**Writing – original draft:** Tereza Štveráková, Jakub Jačisko.

**Writing – review & editing:** Andrew Busch, Alena Kobesová.

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
