## [Decision Letter · Decision Letter 0]

10 May 2021

PONE-D-21-10037

The Impact of COVID-19 on Physical Activity of Czech children

PLOS ONE

Dear Dr. Štveráková,

Thank you for submitting your manuscript to PLOS ONE. After careful consideration, we feel that it has merit but does not fully meet PLOS ONE’s publication criteria as it currently stands. Therefore, we invite you to submit a revised version of the manuscript that addresses the points raised during the review process.

Both reviewers agree that the manuscript is on an interesting topic, but request the authors to better explain the motivation behind their study. In addition to this comment, please also react to the other comments made by the reviewers.

We look forward to receiving your revised manuscript.

Kind regards,

Siew Ann Cheong, Ph.D.

Academic Editor

PLOS ONE

Journal Requirements:

3. Thank you for including your ethics statement: 

"The study was approved by the local institutional ethic committee (EK UK - 173/20).".   

4. In your Methods section, please provide additional information about the participant recruitment method and the demographic details of your participants. Please ensure you have provided sufficient details to replicate the analyses such as: a) the recruitment date range (month and year), b) a description of any inclusion/exclusion criteria that were applied to participant recruitment, c) a table of relevant demographic details, d) a statement as to whether your sample can be considered representative of a larger population, e) a description of how participants were recruited, and f) descriptions of where participants were recruited and where the research took place.

Reviewers' comments:

Reviewer's Responses to Questions

**Comments to the Author**

1. Is the manuscript technically sound, and do the data support the conclusions?

Reviewer #1: Partly

Reviewer #2: Yes

2. Has the statistical analysis been performed appropriately and rigorously? 

Reviewer #1: Yes

Reviewer #2: Yes

3. Have the authors made all data underlying the findings in their manuscript fully available?

Reviewer #1: Yes

Reviewer #2: No

4. Is the manuscript presented in an intelligible fashion and written in standard English?

Reviewer #1: Yes

Reviewer #2: Yes

5. Review Comments to the Author

Reviewer #1: Introduction

The introduction focuses on the history of coronavirus and its spread rather than developing a justification and rationale for the present study.

I would encourage the authors to add to the literature review information on the cultural context. Such as, what are typical rates of physical activity in Czech Republic (i.e. pre-pandemic) among children? How is physical activity typically integrated into the school day in? More information about quarantine restrictions enacted in Czech Republic.

Also, what is the existing literature on the pandemic's impact on children PA? What does this literature tell us and what gaps does this manuscript seek to fill?

Methods

In general the Methods are well described.

The normality test was performed? If yes, please mention.

Please include the effect size.

Table 1 should be in the results section.

Discussion

Lack of theoretical and empirical connection between the variables analysed, as well as, between present findings and previous studies.

L174-186 should be in the introduction section.

Also, I don’t see the need to include statistics values here. They are described in the results section.

References

Ref 42 is already published DOI 10.1016/j.puhe.2020.09.009

Reviewer #2: This in an interesting article dealing with a hot topic. Overall, it is well written and contributes to suggest what prior studies have observed. However, there are a few points that the authors still need to address.

Introduction

-Very little has been said about the situation observed in prior similar studies involving COVID-19 confinements. The bibliography needs to be better revised since several critical studies are missed:

https://pubmed.ncbi.nlm.nih.gov/33271236/

https://pubmed.ncbi.nlm.nih.gov/33311526/

https://pubmed.ncbi.nlm.nih.gov/33733288/

Methods

-I suggest commenting something on your sampling method

-Why did you choose comparing your sample with another different one and don´t do a pre-post design. Kindly explain.

Discussion

-In the limitations, the potential selection bias of your study should be highlighted.

6. PLOS authors have the option to publish the peer review history of their article (what does this mean?). If published, this will include your full peer review and any attached files.

Reviewer #1: No

Reviewer #2: No

---

## [Author Response · Author response to Decision Letter 0]

17 Jun 2021

Dear editor,

We would like to thank you, and the two Reviewers for their thorough reading of the manuscript and highly constructive feedback, which we believe has helped us greatly improve the quality of the manuscript. We also hope the Editor and Reviewers find our revisions responsive to the comments.

Comments and responses

1. 

Thank you for including your ethics statement: 

"The study was approved by the local institutional ethic committee (EK UK - 173/20).". 

Response:

The full name of the ethics committee was added in the Methods section of the manuscript, on page 6, lines 136-138 to read:

The study was approved by the Ethics Committee of the University Hospital Motol and 2nd Faculty of Medicine, Charles University in Prague (EK 1730/20)

Full name of the ethics committee: Ethics Committee of the University Hospital Motol and 2nd Faculty of Medicine, Charles University in Prague. 

Reference No.: EK-1730/20. 

Date of Submission: 15.12.2020

Date of EC Session 6.1.2021

Chairman: MUDr. Vratislav Šmelhaus

2. 

In your Methods section, please provide additional information about the participant recruitment method and the demographic details of your participants. Please ensure you have provided sufficient details to replicate the analyses such as: a) the recruitment date range (month and year), b) a description of any inclusion/exclusion criteria that were applied to participant recruitment, c) a table of relevant demographic details, d) a statement as to whether your sample can be considered representative of a larger population, e) a description of how participants were recruited, and f) descriptions of where participants were recruited and where the research took place.

Response:

The additional information about the participant recruitment method and the demographic details are described on page 6 and 7, lines 141-155 to read:

Our data were collected during COVID lock down from November 2020 to January 2021. It was compared with the norms collected by Cuberek et al. (38) one year earlier, during the same time of the year before the COVID 19 occurred in the Czech Republic. Participants were recruited using an information leaflet created for the purpose of this study. It was published either electronically on an official Dynamic Neuromuscular Stabilization (DNS) website www.rehabps.com or in paper form at physiotherapy centers and at the Motol University Hospital. It contained information on the purpose of the study, a standardized questionnaire and the informed consent. Participation in the study was voluntary and the informed consent was signed by the participant’s parent or legal representative. In total, ninety-eight children participants (56 girls and 42 boys) completed the questionnaire either electronically or as a hard copy. Inclusion criteria for participation comprised age (8-12 years) and participation in distance learning education. Participants were excluded if they had any serious health condition. The questionnaire data revealed participants in this cohort (98), were living in cities and villages of different sizes, came from different elementary schools, and came from various family and social backgrounds. Table 1 shows anthropometric characteristics of the study participants. 

3. 

Please include captions for your Supporting Information files at the end of your manuscript, and update any in-text citations to match accordingly. Please see our Supporting Information guidelines for more information: http://journals.plos.org/plosone/s/supporting-information.

Response:

The supporting information is included at the end of the manuscript on pages 19-20, lines 397-401:

S1 File. Master Data: Pre-COVID and COVID. (ZIP)

S2 File. The Physical Activity Questionnaire for Older Children—The Czech version. (ZIP)

Supplementary Materials: available online at: https://journals.plos.org/plosone/article?id=10.1371/journal.pone.0245256

Reviewer #1: 

1. Introduction

The introduction focuses on the history of coronavirus and its spread rather than developing a justification and rationale for the present study.

I would encourage the authors to add to the literature review information on the cultural context. Such as, what are typical rates of physical activity in Czech Republic (i.e. pre-pandemic) among children? How is physical activity typically integrated into the school day in? More information about quarantine restrictions enacted in Czech Republic.

Also, what is the existing literature on the pandemic's impact on children PA? What does this literature tell us and what gaps does this manuscript seek to fill?

Response:

The introduction section was completely rewritten. Please see introduction section on pages 3-6, lines 49-128.

Well known facts about the pandemic were omitted while the information about PA children in the Czech Republic before the COVID, and Czech Republic COVID restrictions are now described in detail emphasizing that schools were closed for the longest period of time from the whole European Union. COVID restrictions were more strict and long lasting because Czech Republic presented with the second highest confirmed COVID death rate in the world (stated on page 4 lines 91-93). There are two unique aspects about this study. First, this is a study from very seriously COVID affected EU country with the longest period of school closure (declared on page 5, lines 99-101) Second, this is the first study comparing PA of school children during the COVID lockdown with pre-COVID norms defined by the standardized and validated Czech version of questionnaire protocol PAQ-C (PAQ-C/cz). The raw data collected during the same period of year: pre-COVID (Nov 2019-Jan 2020) vs. COVID (Nov 2020- Jan 2021) are compared. This is stated on page 6, lines 126-128.

References to some existing and already very rich literature about pandemic's impact on children PA were added to Introduction section and more recent references were also added in the discussion section.

2. Methods

In general the Methods are well described.

The normality test was performed? If yes, please mention.

Please include the effect size.

Table 1 should be in the results section.

Response:

We did run tests for normality. The total scores for the PAQ-C were normally distributed, as assessed by Shapiro-Wilk's test (p > .05). Scores for each individual question within the PAQ-C were not normally distributed (p < .05). However, we decided to run the test regardless because the independent-samples t-test is fairly robust to deviations from normality. Also, since the total score is generally more telling of an individual's overall activity, we felt non-normality would not substantially affect Type I error rate and the independent-samples t-test can be considered robust.

The Cohen's d effect sizes were included in Table 2. Therefore we reworded the Results section on page 8 and 9, lines 195-207 to read:

Distribution of the 98 participants during COVID lockdown were: males (n = 42, 42.9%), females (n = 56, 57.1%) and the 206 participants included from pre-COVID data were: males (n = 106, 51.5%), females (n = 100, 48.5%). Participant characteristics for both COVID lockdown and pre-COVID data are outlined in Table 1. Not all data was normally distributed, as assessed by Shapiro-Wilk's test. Due to the robustness of the independent samples t-test, data was not altered. Cronbach‘s alpha scores were calculated to score internal consistency for both sets of PAQ-C/cz questionnaire data (pre-COVID and COVID lockdown) using all nine questions. Cronbach’s alpha for pre-COVID questionnaire data (Cuberek et al. (38)) was acceptable at 0.758, and COVID lockdown Cronbach’s alpha was interpreted as good at 0.806 (42). Results of all independent samples t-tests with 95% confidence intervals are presented in Table 2, with gender-specific data presented in Table 3. Significant differences were found in the mean PAQ-C total scores between pre-COVID and COVID lockdown, t(302) = 5.118., p < .001, d = .63, with a mean difference of .385 (95% CI: .237, .532). 

3. Discussion

Lack of theoretical and empirical connection between the variables analyzed, as well as, between present findings and previous studies.

L174-186 should be in the introduction section.

Also, I don’t see the need to include statistics values here. They are described in the results section.

Response:

In the discussion new first paragraphs have been added summarizing the COVID lockdown health aspects and issues reported by recent studies. Page 13. Lines 232-240.

Furthermore, new sections marked in red were added throughout the discussion to compare our results with the results of other authors on the similar topics.

To avoid prolonging the article, the following paragraph was omitted compared to the originally submitted manuscript, as studies on the given topics are already being created or have been published.

Finally, more extensive research on a greater cohort, perhaps during spring and summer season lock downs (if again effective) are needed to generalize the restriction’s effect on the entire pediatric population. Also, future studies should strive to identify any general impact of month-long or year-long sport restrictions have on children’s health including obesity, cardiovascular fitness, back pain, depression and other psychiatric and physical health diseases. Once lock downs are lifted and sport participation can resume, the frequency and severity of sport-related injuries should be explored. It can be expected that reductions in PA are similar for children in different countries with long lasting or repetitive school and sport lock downs. 

Statistic values were deleted from the Discussion section. 

The text that was originally on lines 174-186 was rewritten and this statement “This is the first study comparing PA of school children during the COVID lockdown time with pre-COVID norms defined by the Czech version of PAQ-C (PAQ-C/cz) and by evaluating their number of daily steps” is now in Introduction section on page 6, lines 126-128.

4. References

Ref 42 is already published DOI 10.1016/j.puhe.2020.09.009

Response: The reference was updated:

29. Pombo A, Luz C, Rodrigues LP, Ferreira C, Cordovil R. Correlates of children’s physical activity during the COVID-19 confinement in Portugal. Public Health. 2020 Dec;189:14–9. 

Reviewer #2: 

This in an interesting article dealing with a hot topic. Overall, it is well written and contributes to suggest what prior studies have observed. However, there are a few points that the authors still need to address.

1. Introduction

Very little has been said about the situation observed in prior similar studies involving COVID-19 confinements. The bibliography needs to be better revised since several critical studies are missed:

https://pubmed.ncbi.nlm.nih.gov/33271236/

https://pubmed.ncbi.nlm.nih.gov/33311526/

https://pubmed.ncbi.nlm.nih.gov/33733288/

Response:

Following the first reviewer suggestions the introductions section was completely rewritten explaining pre-pandemic children’s PA in Czech Republic, reporting the level of PA integrated in Czech schools and more details about quarantine restrictions enacted in Czech Republic are explained. 

The existing literature on the pandemic's impact on children PA is summarized including the suggested references:

Reference https://pubmed.ncbi.nlm.nih.gov/33271236/ Ref No. 11

Reference https://pubmed.ncbi.nlm.nih.gov/33311526/ Ref No. 26

Reference https://pubmed.ncbi.nlm.nih.gov/33733288/ Ref. No. 20

2. Methods

a) I suggest commenting something on your sampling method

b) Why did you choose comparing your sample with another different one and don´t do a pre-post design. Kindly explain.

Response:

a) Comments on sampling method are now described in lines 141-155. 

b) Since COVID-19 broke out in March 2020 and our study was designed in July that year, we did not have time to collect our own pre-COVID data. We had no idea that a pandemic would break out and some pre-COVID data would be needed. That’s why we were searching for an adequate study design we could follow. We found Cuberek et al. study (Ref No 38) dealing with the standardization of the PAQ-C questionnaire for Czech children. For the purposes of this study data were collected from 206 children. The authors provided us the data and on the basis of them we performed a comparison with our COVID data in the same time period (November-January 2019-2020 vs. 2020-2021) which makes this study unique. This is stated in Methods section on page 6, lines 141-143 and in Discussion section on page 17, lines 330-332.

3. Discussion

In the limitations, the potential selection bias of your study should be highlighted.

Response:

Selection bias is now explained on page 19, lines 381-383 to read:

Another limitation could be the fact that the study may not fully represent the population as a whole. Parents who are not upset by the lockdown are perhaps not as motivated to complete a questionnaire regarding their child’s lack of PA.

Funding, page 20, lines 419-421: 

This study was supported by the foundation Movement without Help, Prague, Czech Republic, by Rehabilitation Prague School www.rehabps.com and by Institutional research program Progres Q41.

---

## [Editor Report · Decision Letter 1]

23 Jun 2021

The Impact of COVID-19 on Physical Activity of Czech children

PONE-D-21-10037R1

Dear Dr. Štveráková,

We’re pleased to inform you that your manuscript has been judged scientifically suitable for publication and will be formally accepted for publication once it meets all outstanding technical requirements.

Kind regards,

Siew Ann Cheong, Ph.D.

Academic Editor

PLOS ONE
---

## [Editor Report · Acceptance letter]

28 Jun 2021

PONE-D-21-10037R1 

The Impact of COVID-19 on Physical Activity of Czech children 

Dear Dr. Štveráková:

I'm pleased to inform you that your manuscript has been deemed suitable for publication in PLOS ONE. Congratulations! Your manuscript is now with our production department. 

Kind regards, 

on behalf of

Dr. Siew Ann Cheong 

Academic Editor

PLOS ONE